# Factors Influencing Depression in Adolescents Focusing on the Degree of Appearance Stress

Mihye Lim [1] and Myoungjin Kwon [2,*]

1   Department of Nursing, Joongbu University, Chungnam 32713, Republic of Korea
2   Department of Nursing, Daejeon University, Daejeon 34520, Republic of Korea
*   Correspondence: mjkwon@dju.kr; Tel.: +82-42-820-2665

**Abstract:** This descriptive cross-sectional study examined the factors that affect depression in adolescents focusing on how stressed they are regarding their appearance (hereafter, degree of appearance stress). Data from 6493 adolescents from the 2020 Korean Youth Risk Behavior Survey were used. Using SPSS 25.0, a complex sample plan file was created, weighted, and analyzed. The frequency, chi-square test, independent t-test, and linear regression were used for the complex sample analysis. The results showed that among adolescents with low appearance stress, depression was significantly affected by the number of breakfast meals, weight control efforts, smoking, loneliness, subjective physical appearance, and smartphone overdependence. Among those with high appearance stress, depression was significantly affected by academic grades, weight control efforts, drinking habits, loneliness, subjective physical appearance, and smartphone overdependence. Furthermore, these factors differed according to the degree of appearance stress. Thus, while developing interventions for mitigating depression in adolescents, the degree of appearance stress should be considered, and a differentiated strategy should be used accordingly.

**Keywords:** adolescents; depression; regression; stress; survey

## 1. Introduction

COVID-19 has drastically affected society. For instance, in Korea, school reopening has been postponed several times because of social distancing, and classes have been completely online. These environmental changes have reportedly increased depressive tendencies in adolescents due to the limitations in social activities and lack of social interaction [1–3].

Depression has been the most common mental illness among adolescents aged 10–19 in South Korea since 2018. Data from the Health Insurance Review and Assessment Service revealed that the number of depressed patients aged 10–19 increased by approximately 40,000 in 2020 compared to 2016 [4]. Depression in adolescence affects delinquent behavior, school dropout, suicidal ideation, and suicidal impulses. It is also a major predictor of depression in adulthood [5–7]. In addition, experiencing depression in adolescence is associated with a higher rate of emergency room visits or hospitalization than experiencing it in adulthood depression; if not treated, the rate of suicide attempts in adulthood increases by approximately four times [8,9]. Therefore, the early detection of depression in adolescents and its treatment and management through appropriate interventions are crucial [10].

Adolescence is a period during which individuals experience various physical, social, and mental changes along the continuum of human development [11]. Moreover, adolescents are particularly interested in their appearance because they experience rapid physical changes [11]. They are continuously and unconsciously exposed to unrealistic beauty standards, appearances, or body shapes in mass media programs or games and are affected by the formation of individual body images [12]. Notably, if they believe they do not conform to these superficial and seemingly unrealistic beauty standards, their self-esteem decreases. In addition, they are sensitive to others' judgement. They desire

positive recognition from others, and their sensitivity to criticism increases [6]. In female adolescents, being judged as sexual objects leads to the internalization of the sociocultural value of appearance, which increases depression through the monitoring of one's own body [11]. In other words, physical judgment is internalized. Through this, one's body is continuously observed and controlled for the perspectives of others, thereby increasing depression. Therefore, it is necessary to consider the relationship between depression in adolescents and the degree of appearance stress.

The research on the relationship between appearance and depression in multicultural adolescents shows that even if adolescents are bullied, satisfaction with appearance can reduce depression and improve academic achievement [13]. Regarding the effect of narcissism on depression, research has shown that the degree of dissatisfaction with appearance influences depression [6]. Notably, appearance stress enhances depression and reduces self-esteem [12]. Therefore, the appearance stress of adolescents affects depression, although there are other factors affecting depression as well. However, to the best of our knowledge, few studies have examined the factors affecting depression according to the degree of appearance stress among adolescents. Therefore, this study examined the factors affecting depression in adolescents, focusing on the degree of appearance stress. Understanding these aspects can yield insight into the design and implementation of appropriate intervention programs for the management of depression in adolescents.

## 2. Materials and Methods

### 2.1. Study Design

This study conducted a secondary analysis of data from the first year of the 2020 Korean Youth Risk Behavior Survey (KYRBS). A descriptive research study was conducted to understand the factors affecting depression in adolescents focusing on the degree of appearance stress.

### 2.2. Participants

The 2020 KYRBS is a survey jointly conducted by the Korea Centers for Disease Control and Prevention and the Ministry of Education to identify the health behaviors of Korean adolescents. A total of 54,948 individuals aged 12–18 years were surveyed in 2020. The 2020 KYRBS asked the following question regarding stress: "How much stress do you normally feel?" The responses were recorded using a five-point Likert scale, indicating either "I feel very much", "I feel a lot", "I feel a little", "I do not feel much", or "I do not feel at all". Of these, only 6493 respondents answered that the cause of their stress was their appearance; these respondents were selected as subjects for this study. Those who responded "I do not feel it at all" were excluded. Subsequently, the selected participants were divided into two groups based on their degree of appearance stress. The low-appearance stress group, which included those who had answered "I feel a little" or "I do not feel much" had 4524 respondents. Meanwhile, the high-appearance stress group, which included those who responded "I feel very much" or "I feel a lot", had 1969 respondents.

### 2.3. Study Variables

The study variables drawn from the KYRBS items were classified into four types: sociodemographic, diet-related, physical health, and psychological factors.

The sociodemographic factors included sex (male or female), age, academic grade (upper, middle, or lower), and whether they lived with family (yes or no).

The diet-related factors included the number of breakfasts over the last seven days (0, 1–3, or 4–7), instances of fast food intake over the last seven days (0, 1–2/week, 3–6/week, or more than once daily), and nutrition and dietary education (yes or no).

The physical health factors comprised weight control efforts over the last 30 days (decreased effort, increased and maintained effort, or no effort), smoking (yes or no), drinking (yes or no), fatigue recovery by sleep over the last seven days (enough or not

enough), and body mass index (BMI) (less than 18.5 kg/m$^2$, 18.5–22.9 kg/m$^2$, or 23 kg/m$^2$ or more) [14].

The psychological factors included depression (yes or no), loneliness (feeling less or feeling more), subjective health (good, normal, or bad), subjective physical appearance (underweight, normal, or overweight), smartphone overdependence (yes or no), and anxiety. Smartphone addiction was measured by the Ministry of Science, ICT, and Future Planning and the Korea Information Society Agency [15] using the smartphone dependence scale, which integrates the existing internet (K-scale) and smartphone (S-scale) individual scales. This is a four-point Likert scale with a total of 10 items, with scores ranging from 10 to 40. Among adolescents, 31 points or more correspond to the high-risk group, whereas 23–30 points correspond to the potential-risk group. Therefore, 23 was set as the threshold for smartphone addiction. Anxiety was assessed using the Generalized Anxiety Disorder-7 (GAD-7) scale, developed by Spitzer et al. [16]. The GAD-7 comprises seven items rated on a four-point Likert scale; it measures the frequency of experiencing the symptoms of anxiety over the past two weeks. The scores range from 0 to 21, with higher scores indicating more severe anxiety.

*2.4. Statistical Analysis*

The collected data were analyzed after generating a complex sample plan file by assigning weights using SPSS 25.0. The significance level was set at *p* < 0.05. The complex samples chi-square test and an independent *t*-test were then used to compare the weighted percentage or mean of all variables with respect to the two groups corresponding to the different degrees of appearance stress. Finally, the variables with a significant association in the chi-square test and independent *t*-test were analyzed using logistic regression analysis to examine the association between depression and these variables.

**3. Results**

*3.1. Comparison of Sociodemographic Factors*

The results in Table 1 regarding the sociodemographic factors show that sex, age, and academic grade differed significantly between the two groups. Specifically, women and those aged 16–18 years were more stressed about their appearance.

**Table 1.** Sociodemographic factors (*n* = 6493).

| Characteristics | | Low Appearance Stress (*n* = 4524) | High Appearance Stress (*n* = 1969) | x$^2$ (p) |
|---|---|---|---|---|
| | | *n* (Weight%) | | |
| Sex | Male | 2038 (44.9) | 597 (31.0) | 112.28 (<0.001) |
| | Female | 2486 (55.1) | 1372 (69.0) | |
| Age (year) | 12–15 | 3238 (69.9) | 1351 (66.5) | 7.58 (0.017) |
| | 16–18 | 1283 (30.1) | 613 (33.5) | |
| Academic grade | Upper | 1512 (33.8) | 532 (27.4) | 63.45 (<0.001) |
| | Middle | 1364 (30.0) | 518 (25.9) | |
| | Lower | 1648 (36.2) | 919 (46.7) | |
| | No | 3234 (71.9) | 1210 (62.2) | |
| Living with family | Yes | 4373 (97.5) | 1886 (96.7) | 2.64 (0.094) |
| | No | 151 (2.5) | 83 (3.3) | |

*3.2. Diet-Related Factors*

Table 2 shows the results for the diet-related factors. The two groups differed significantly in the number of breakfasts per week, fast food intake per week, and nutritional

education ($p < 0.05$). Specifically, the low-appearance stress group ate breakfast more often per week and had more instances of fast-food intake per week. Nutritional education was also higher in the low-appearance-related stress group.

**Table 2.** Diet-related factors ($n$ = 6493).

| Characteristics | | Low Appearance Stress ($n$ = 4524) | High Appearance Stress ($n$ = 1969) | $x^2$ ($p$) |
|---|---|---|---|---|
| | | $n$ (Weight%) | | |
| Number of breakfasts per week | 0 | 941 (20.6) | 562 (27.9) | 78.80 (<0.001) |
| | 1–3 | 1247 (27.2) | 634 (31.6) | |
| | ≥4 | 2336 (52.1) | 773 (40.5) | |
| Instances of fast-food intake per week | 0 | 352 (18.3) | 816 (18.0) | 14.54 (0.002) |
| | 1–4 | 1485 (75.1) | 3515 (77.7) | |
| | ≥5 | 132 (6.6) | 193 (4.3) | |
| Nutrition education | Yes | 2238 (48.8) | 907 (44.6) | 9.87 (0.004) |
| | No | 2286 (51.2) | 1062 (55.4) | |

### 3.3. Comparison of Health Factors

Table 3 shows the results for the health factors. The two groups showed significant differences in all physical factors ($p < 0.05$), except for BMI. Specifically, the high-appearance stress group made more efforts to lose weight, smoked, and drank more. Fatigue recovery during sleep was higher in the low-appearance stress group.

**Table 3.** Health related factors ($n$ = 6493).

| Characteristics | | Low Appearance Stress ($n$ = 4524) | High Appearance Stress ($n$ = 1969) | $x^2$ ($p$) |
|---|---|---|---|---|
| | | $n$ (Weight%) | | |
| Weight control effort | Reduction effort | 2271 (50.6) | 1160 (58.3) | 34.67 (<0.001) |
| | Increase, maintenance effort | 871 (18.9) | 328 (16.8) | |
| | No effort | 1382 (30.5) | 481 (24.9) | |
| Smoking | Yes | 494 (11.0) | 300 (14.5) | 15.16 (<0.001) |
| | No | 4030 (89.0) | 1669 (85.5) | |
| Drinking | Yes | 1484 (32.5) | 844 (42.5) | 59.82 (<0.001) |
| | No | 3040 (67.5) | 1125 (57.5) | |
| Fatigue recovery by sleep | Much | 1657 (36.8) | 379 (19.5) | 200.88 (<0.001) |
| | A little | 2867 (63.2) | 1590 (80.5) | |
| Body mass index (kg/m$^2$) | <18.5 | 800 (18.6) | 297 (16.6) | 4.50 (0.150) |
| | 18.5–22.9 | 2008 (47.3) | 828 (47.2) | |
| | ≥23 | 1483 (34.1) | 673 (36.2) | |

### 3.4. Comparison of Psychological Factors

Table 4 shows that all psychological factors differed between the two groups ($p < 0.05$). Depression and loneliness appeared more frequently in subjects with high appearance

stress. The low- (high-) appearance stress group was subjectively perceived as healthier (obese). Both smartphone overdependence and anxiety were more prominent in the high-appearance stress group.

**Table 4.** Psychological factors (*n* = 6493).

| Characteristics | | Low Appearance Stress (*n* = 4524) | High Appearance Stress (*n* = 1969) | $x^2$/t (*p*) |
|---|---|---|---|---|
| | | *n* (Weight%)/Mean (SE) | | |
| Depression | Yes | 727 (16.2) | 945 (48.2) | 696.28 (<0.001) |
| | No | 3797 (83.8) | 1024 (51.8) | |
| Loneliness | Feeling less | 4184 (92.4) | 1256 (64.1) | 743.96 (<0.001) |
| | Feeling much | 340 (7.6) | 713 (35.9) | |
| Subjective health | Good | 3410 (75.4) | 1088 (54.1) | |
| | Normal | 928 (20.2) | 598 (30.6) | 348.35 (<0.001) |
| | Bad | 186 (4.3) | 283 (15.3) | |
| Subjective physical appearance | Underweight | 849 (18.8) | 319 (16.4) | |
| | Normal | 1442 (31.4) | 479 (24.6) | 49.16 (<0.001) |
| | Overweight | 2233 (49.8) | 1171 (59.0) | |
| Smartphone overdependence | Yes | 11074 (25.0) | 768 (39.7) | 138.30 (<0.001) |
| | No | 3417 (75.0) | 1201 (60.3) | |
| Anxiety | | 2.61 (0.05) | 6.97 (0.13) | 31.92 (<0.001) |

### 3.5. Factors Related to Depression

Logistic regression analyses were performed using all variables as independent variables, except for living with family and BMI, which did not show significant differences in the results of the different analyses. Depression was used as the dependent variable. The results are summarized in Table 5.

In the low-appearance stress group, the number of breakfasts per week, weight control efforts, smoking, loneliness, subjective body appearance, and anxiety significantly affected depression. The odds of being depressed was 1.40 times higher for subjects who did not eat breakfast (95% CI: 1.08–1.81) and 1.43 times higher for subjects who tried to gain or maintain weight (95% CI: 1.07–1.91). The odds of being depressed was 1.56 times higher for those who smoked (95% CI: 0.46–0.90) and 2.85 times higher for those who experienced greater loneliness (95% CI: 0.26–0.48). The odds of being depressed were 1.35 and 1.32 times higher in subjects who perceived their subjective body type as underweight and normal, respectively, than those who considered themselves overweight (95% CI: 1.01–1.82 and 1.03–1.68, respectively). Finally, the odds of being depressed were 1.18 times higher as anxiety increased (95% CI: 1.14–1.22).

In the high-appearance stress group, academic performance, weight control efforts, drinking, loneliness, subjective body appearance, and anxiety were significantly associated with depression. The odds of being depressed were 1.58 times lower in adolescents with poor academic performance (95% CI: 0.47–0.84). The odds of being depressed were 1.40 (95% CI: 1.04–1.88) and 1.59 (95% CI: 1.09–2.32) times higher in adolescents who tried to lose weight or increased or maintained their weight, respectively, than those who made efforts to control their weight. The odds of being depressed were 1.49 times higher in drinkers (95% CI: 0.53–0.86) and 1.85 times higher in those who were frequently lonely (95% CI: 0.42–0.70). The participants who perceived their subjective body shape as normal had a 1.53-fold higher level of depression (95% CI: 1.18–1.98). Finally, as anxiety increased, the likelihood of depression increased (95% CI: 1.09–1.16).

Table 5. Factors related to depression (*n* = 6493).

| Characteristics | Categories | Low Appearance Stress (*n* = 4524) | | | High Appearance Stress (*n* = 1969) | | |
|---|---|---|---|---|---|---|---|
| | | OR | 95% CI | *p* | OR | 95% CI | *p* |
| Sex (ref: female) | Male | 0.82 | 0.64–1.03 | 0.099 | 0.87 | 0.66–1.13 | 0.306 |
| Age (ref: 16–18) | 12–15 | 0.94 | 0.74–1.20 | 0.641 | 1.13 | 0.89–1.45 | 0.296 |
| Academic grade (ref: lower) | Upper | 0.87 | 0.64–1.12 | 0.301 | 0.63 | 0.47–0.84 | 0.002 |
| | Middle | 1.01 | 0.78–1.28 | 0.984 | 0.87 | 0.66–1.14 | 0.325 |
| Number of breakfasts (ref: ≥4) | 0 | 1.40 | 1.08–1.81 | 0.010 | 1.14 | 0.85–1.53 | 0.355 |
| | 1–3 | 1.15 | 0.90–1.47 | 0.259 | 1.28 | 0.98–1.67 | 0.069 |
| Instances of fast food intake (ref: ≥5) | 0 | 0.81 | 0.47–1.37 | 0.436 | 0.70 | 0.39–1.28 | 0.254 |
| | 1–4 | 0.83 | 0.51–1.36 | 0.474 | 0.86 | 0.50–1.46 | 0.586 |
| Nutrition education (ref: yes) | No | 0.84 | 0.68–1.04 | 0.128 | 0.92 | 0.74–1.16 | 0.523 |
| Weight control effort (ref: no effort) | Reduction effort | 1.30 | 0.99–1.70 | 0.053 | 1.40 | 1.04–1.88 | 0.026 |
| | Increase, maintenance effort | 1.43 | 1.07–1.91 | 0.013 | 1.59 | 1.09–2.32 | 0.016 |
| Smoking (ref: yes) | No | 0.64 | 0.46–0.90 | 0.011 | 1.12 | 0.77–1.62 | 0.539 |
| Drinking (ref: yes) | No | 0.78 | 0.61–1.01 | 0.058 | 0.67 | 0.53–0.86 | 0.002 |
| Fatigue recovery by sleep (ref: a little) | Much | 0.82 | 0.65–1.04 | 0.104 | 0.98 | 0.73–1.31 | 0.906 |
| Loneliness (ref: feeling much) | Feeling less | 0.35 | 0.26–0.48 | <0.001 | 0.54 | 0.42–0.70 | <0.001 |
| Subjective health (ref: bad) | Good | 1.25 | 0.77–2.02 | 0.358 | 1.18 | 0.83–1.68 | 0.332 |
| | Normal | 1.32 | 0.80–2.18 | 0.272 | 1.22 | 0.85–1.75 | 0.268 |
| Subjective physical appearance (ref: overweight) | Underweight | 1.35 | 1.01–1.82 | 0.048 | 1.02 | 0.71–1.45 | 0.906 |
| | Normal | 1.32 | 1.03–1.68 | 0.024 | 1.53 | 1.18–1.98 | 0.001 |
| Smartphone overdependence (ref: yes) | No | 0.88 | 0.71–1.10 | 0.285 | 0.91 | 0.71–1.15 | 0.454 |
| Anxiety | | 1.18 | 1.14–1.22 | <0.001 | 1.12 | 1.09–1.16 | <0.001 |

## 4. Discussion

This study investigated factors affecting the prevalence of depression in adolescents focusing on their appearance stress using the 2020 KYRBS data. The results showed that the ratios of female adolescents, those between 16 and 18 years of age, and those who had lower academic achievement were higher in the high-appearance stress group than in the low-appearance stress group. During adolescence, people react sensitively to changes in their bodies and pay substantial attention to their appearance; women reportedly perceive themselves more negatively than men [11]. Thus, one can argue that women are more stressed by their appearance. Research has shown that interest in appearance increases after establishing sexual identity following secondary growth [12]. The findings of the present study corroborate this; many respondents with high appearance stress were older. Regarding academic achievement, the present results are consistent with those of Oh and Min's [13]: low appearance satisfaction increases depression and reduces academic achievement. Overall, these results show that interventions to reinforce a positive physical self-image is needed in adolescence, and differentiated strategies that consider sex and age should be adopted.

Regarding diet-related differences in the groups, the high-appearance stress group ate breakfast and fast food less often. In terms of health-related group differences, the high-appearance stress group made more effort to lose weight but also smoked and drank

more. They also had fewer days of restorative sleep and engaged in less physical activity. However, no differences were observed in their BMI, probably because adolescents with high appearance stress better manage their weight to appear slimmer and more attractive. Consequently, their dietary intake may be irregular, and they may eat less food. A survey of adolescents showed that those with a high propensity for appearancism had lower body satisfaction and tended to control their meals more, resulting in a lower frequency of fast food and snack intake. Moreover, their physical activity levels were lower, consistent with the results of this study [16]. This is because adolescents pay attention to how they are viewed by others and are interested in how their physical characteristics appear to others. Therefore, idealism is observed because individuals want to make their bodies attractive to people around themselves [17]. Notably, few studies have explained the relationship between nutritional education and appearance stress, which is a gap addressed by this study. Adolescence is a period of rapid growth; possessing the right nutritional knowledge and eating a balanced diet during this period are crucial for forming the correct body image and reaching healthy adulthood. Therefore, considering adolescents' degree of appearance stress and knowledge about nutrition, psychological support should be provided together with nutritional education so that adolescents can receive balanced nutrition.

Regarding psychological characteristics, the high-appearance stress group experienced more depression and loneliness than the low-appearance stress group. Furthermore, more adolescents in the former believed their subjective health condition was poor, perceived themselves as overweight or underweight, responded that they overused their smartphones, and reported higher anxiety scores. Thus, the high-appearance stress group showed more psychological problems than the low-appearance stress group. In general, stress affects depression [18]. In line with the findings of this study, research has shown that appearance is related to depression [11]. Health status-related cognition directly and indirectly affects stress and depression in adolescents [18], and underestimating or overestimating one's body image affects appearance stress and anxiety [19]. Previous research has shown that judgments about appearance contribute to psychological and biological stress and influence depression [20]. Meanwhile, if appearance stress is high because awareness of others is excessively high, then one may continuously try to improve one's appearance, feel lonely or anxious, or stay away from others due to the lack of self-confidence. However, only a few studies have explored this. Further qualitative explorations of adolescents' experiences of appearance stress are needed. Researchers and practitioners should investigate interventions that can help resolve the psychological problems of adolescents experiencing appearance stress.

Regardless of the degree of appearance stress, loneliness and anxiety affect depression in adolescents. A meta-analysis showed that loneliness affects depression [21] and is a key symptom of adolescent depression [22]. Adolescent loneliness refers to the absence of relationships or social interactions [23]. In severe cases, it can lead to atrophy or frustration in relationships, problematic or addictive behaviors, or suicidal impulses [24]. Moreover, adolescent anxiety is known to be highly correlated with depression because they overlap structurally; therefore, anxiety often appears simultaneously with depression [25]. Developing and implementing a depression management program that considers adolescent loneliness and anxiety is crucial for mitigating mental and emotional problems.

Additionally, in the low-appearance stress group, depression was higher in the group that responded that they were underweight or normal weight compared to the group that did not eat breakfast at all, tried to gain weight, smoked, or were overweight. Parenting behavior can substantially influence changes in adolescent depression [26]. For adolescents, breakfast is often served by others or caregivers. Therefore, whether adolescents eat breakfast may be related to their caregivers' parenting behaviors. The findings of this study on smoking are also in line with those of previous studies. For instance, a study of multicultural adolescents found that smoking influenced depression [27]. However, because this was a cross-sectional study, it was difficult to determine whether people were depressed because of smoking or whether they smoked because they were depressed. Therefore, lon-

gitudinal studies should be conducted to explore the causal relationship between smoking and depression. Next, those who made efforts to gain weight and responded that their appearance was underweight or average showed higher levels of depression, consistent with prior research [19]. Therefore, when developing a program to prevent and manage depression in adolescents, complex factors, including their perception of their subjective body shape, should be considered, even if they have low appearance stress.

The high-appearance stress group also had a low academic performance. Furthermore, those who undertook weight control efforts experienced high levels of depression. Compared to those who drank and were overweight, depression was higher in those who responded normally. A study targeting multicultural adolescents found that appearance stress influenced depression and that depression affected academic performance [13]. Appearance stress during adolescence is often related to adolescent self-esteem [12], which is also related to academic achievement. High self-esteem among adolescents is also associated with high academic achievement [28]. Therefore, when adolescents experience high appearance stress, their self-esteem may decrease. This, in turn, can lead to lower academic achievement and depression. However, few studies have conducted path analyses of academic achievement, appearance stress, and depression. Future studies should explore this among adolescents. In the high-appearance stress group, those who made efforts to control their weight and responded that their body shape was normal showed high levels of depression. This may be due to the fact of negative feedback, which causes them more stress. A previous study found depression to be high in those who made all kinds of weight control efforts, including exercise [29]. If an individual is overweight, they need to control their weight to prevent diseases in adulthood or form a healthy body image. However, in the case of weight control to maintain an ideal body shape, adolescents should be made aware that excessive worry about weight control can cause depression. Efforts are needed to help adolescents develop a healthier body image.

Finally, this study was a secondary analysis using the 2020 KYRBS data. Due to the cross-sectional nature of the data, it was difficult to establish causal relationships. Nevertheless, it is meaningful to explore the factors influencing depression in relation to appearance using big data because this is a recent and important problem for adolescents.

## 5. Conclusions

This study used data from the 2020 Korean Youth Risk Behavior Survey to determine the factors affecting depression in adolescents focusing on the degree of appearance stress. This study found that in adolescents with low appearance stress, the number of breakfasts, weight control efforts, smoking, loneliness, subjective physical appearance, and anxiety had significant effects on depression. However, for those with high appearance stress, academic performance, weight control efforts, drinking, loneliness, subjective appearance, and anxiety were significantly associated with depression. These factors differed according to their degree of appearance stress. Therefore, when developing interventions to improve depression in adolescents, it is necessary to consider the degree of appearance stress and use different strategies accordingly. Thus, further studies on adolescent appearances are warranted.

**Author Contributions:** Conceptualization, M.L. and M.K.; methodology, M.L. and M.K.; software, M.K.; data curation, M.L. and M.K.; writing—original draft preparation, M.L. and M.K.; writing—review and editing, M.L. and M.K.; visualization, M.L.; supervision, M.K. All authors have read and agreed to the published version of the manuscript.

**Funding:** This study received no external funding.

**Institutional Review Board Statement:** Not applicable.

**Informed Consent Statement:** Informed consent was obtained from all the participants involved in the study.

**Data Availability Statement:** Publicly available datasets were analyzed in this study. These data can be found at https://www.kdca.go.kr/yhs (accessed 25 September 2022).

**Conflicts of Interest:** The authors declare no conflict of interest.

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
