# Peer review of "Factors Influencing Depression in Adolescents Focusing on the Degree of Appearance Stress"

_nursrep, doi:10.3390/nursrep13010047_

Round 1
Reviewer 1 Report
General:
The study uses the 2020 Korean Youth Risk Behavior Survey (KYRBS) data on Korean adolescents to examine the degree of appearance stress in adolescents and which factors affect depression in them according to the degree of adolescent appearance stress [lines: 67-70].
The study used statistical methods such as the Chi 2 test of independence and logistic regression analysis [lines: 121-127].
Detailed comments:
[Lines: 163-164] table 5 shows that the result is 1.40 times higher for the number of breakfasts equal to 0, and not less than or equal to 3. For the number less than or equal to three, the result is statistically insignificant p=0.259.
[Lines: 163 and more] The chance of becoming depressed is 1.4 times higher, not being depressed is 1.4 times higher. Similarly for subsequent comparisons.
[Lines: 73-127] The division of subchapter 2.3 into subchapters seems unjustified. Individual subsections contain little content. It seems that the individual factors should be described in more detail.
Summary:
The study is based on the analysis of data obtained from KYRBS.
Very modestly described research methodology that should be completed [lines: 73-127]. It is worth pointing out why such research methods were chosen. Why were these types of statistical tests chosen over others? Why was logistic regression chosen over other types? Why were these types of factors affecting depression chosen? No clearly posed research question.
In logistic regression models, only one factor is analyzed at a time, the question is what would the model look like if multiple regression was used to account for multiple factors at once [lines: 154-180].
No comparisons with test results in countries other than Korea.
The conclusions section is very sketchy and needs to be completed [lines: 286-295].
Relatively few literature sources, mostly Korean authors [lines: 307-366].
Author Response
Thank you for your careful review. The answer to the review has been attached as a file.

Reviewer 2 Report
The object of study seems to me very interesting and timely, given the high levels of mental health that affect adolescents and young people throughout the world. It is a topic that needs a lot of research and knowledge. That is why I believe that this work contributes, even if it is a study based on secondary sources, to improve the knowledge of a complex psychosocial problem. The article, as noted in the study itself, fills the gap in some aspects (nutritional education, for example) and offers ideas for carrying out this type of study in other countries. The use of this type of secondary data sources is a great opportunity to improve knowledge in these fields of specialization, and above all, if analyzes of variables and factors that can be related to the reality of mental health are carried out. The problem of depression in adolescents is a crucial phenomenon, but addressing the issue of physical appearance in those years is crucial to find solutions to a problem that is growing considerably. Therefore, the paper presents a relevant social and scientific object of study. The introduction and conceptual justification of the problem seems to me correct for this type of study, above all I believe that it does not need to broaden the context or the conceptualization to understand the relevance of the topic to be dealt with by the authors. The methodology seems timely, correct and rigorous. The data analyzes are rigorous and relevant. The Discussion section could have further increased the implications of the results, but it has discussed well with the other authors and has indicated what measures could be taken to improve the situation. I think it would have been good to broaden the gender perspective to improve the interpretation of the different results according to the sexes. But the type of study offers an overview of the subject and does so in a solvent way. For all of this, I think it is a very interesting and pertinent paper that should be accepted for publication.
Author Response
Thank you for your detailed and kind review. In the next study, we will also analyze gender differences. Thank you again.
Reviewer 3 Report
I have read the article entitled "Factors influencing depression in adolescents according to the degree of appearance stress". This is a very interesting study, but its quality will be improved by adding a few more elements. Unfortunately, there is no information on the approval of the Bioethical Committee for the study, both in the text of the manuscript and in the final information. It is also worth supplementing the content of the article with a description of the limitations of the presented article. In addition, I also suggest expanding the summary section, which, in particular with the discussion section, is disproportionately shorter.
Author Response
Thank you for your kind review.
The KYRBS (Korean Youth Risk Behavior Survey) is a survey conducted jointly by the Korea Centers for Disease Control and Prevention, and the Ministry of Education to identify the health behaviors of Korean adolescents. The KYRBS received the Korea Centers for Disease Control and Prevention (KCDC) IRB approval (2014-06EXP-02-P-A) in 2014. From 2015, the ethics approval for the KYRBS was waived by the KCDC IRB under the Bioethics & Safety Act and opened to the public for academic use. And the conclusion has been revised overall.
Reviewer 4 Report
The authors present a well prepared and presented paper, the analysis is pertinent and the results are clearly presented. It remains to be clarified whether the sample size does not affect the significance of the chi square test (due to the sensitivity of the latter to large sample sizes). Logistic regression is a good proposal for analysis; however, it does not estimate the covariance of the factors that predict the low self-esteem of the respondents; perhaps an alternative analysis suggested for future work would be to use structural equation modeling.
Author Response
For the analysis of the data, a composite sample analysis was performed as recommended by the Korea Centers for Disease Control and Prevention. In the next study, we will consider applying structural equation modeling. Thank you
Round 2
Reviewer 1 Report
Thank you for sending your response to the previous review and the changes made.
Regarding the proposed changes in the first review:
1. proposed changes were made regarding [Lines: 163-164 (now - lines: 159-160)]
2. made proposed changes regarding [Lines: 163 and more (now - lines: 159 and more)]
3 Removed the division of Chapter 2.3 as proposed [Lines 73-127 (now - lines: 72-123)]
4. the methodology of the study still seems very vaguely described, however, an explanation is indicated in the response
5 Comparisons are made mainly with Korean studies, but it should be noted that one additional literature item from outside Korea has been added
6. the summary chapter was completed by which it became better suited to the structure of the work. It could still be worked on, but it is acceptable
7. relatively few literature items by authors from outside Korea - however, the authors indicate in their response to the review that this type of research is mainly conducted in Korea - it would be worth mentioning this explicitly in the article
Necessary changes:
[Line: 225] The literature reference should be to item 20.
[Line: 227] The literature reference should be to item 19. This is a newly added item.
[Lines: 291] For adolescents with low stress, anxiety is significant (p<0.001) and not smartphone overdependence (p=0.285).
[Lines: 294] For adolescents with high stress, anxiety is significant (p<0.001) and not smartphone overdependence (p=0.454).
Author Response
Thank you for your careful review.
[Line: 225] The literature reference should be to item 20.
Answer: Reference No. 19 has been modified to No. 20.
[Line: 227] The literature reference should be to item 19. This is a newly added item.
Answer: Reference No. 19 has been modified to No. 20.
[Lines: 291] For adolescents with low stress, anxiety is significant (p<0.001) and not smartphone overdependence (p=0.285).
Answer: Deleted smartphone overdependence and added anxiety.
[Lines: 294] For adolescents with high stress, anxiety is significant (p<0.001) and not smartphone overdependence (p=0.454).
Answer: Deleted smartphone overdependence and added anxiety.